# Digital Twins in Solar Farms: An Approach through Time Series and Deep Learning

**Kamel Arafet \* and Rafael Berlanga** 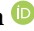

Department of LSI, Campus de Ríu Sec, Universitat Jaume I, E-12071 Castelló de la Plana, Spain; berlanga@uji.es
\* Correspondence: al393990@uji.es

**Abstract:** The generation of electricity through renewable energy sources increases every day, with solar energy being one of the fastest-growing. The emergence of information technologies such as Digital Twins (DT) in the field of the Internet of Things and Industry 4.0 allows a substantial development in automatic diagnostic systems. The objective of this work is to obtain the DT of a Photovoltaic Solar Farm (PVSF) with a deep-learning (DL) approach. To build such a DT, sensor-based time series are properly analyzed and processed. The resulting data are used to train a DL model (e.g., autoencoders) in order to detect anomalies of the physical system in its DT. Results show a reconstruction error around 0.1, a recall score of 0.92 and an Area Under Curve (AUC) of 0.97. Therefore, this paper demonstrates that the DT can reproduce the behavior as well as detect efficiently anomalies of the physical system.

**Keywords:** time series; autoencoders; digital twins; photovoltaic solar farm; Internet of Things; Industry 4.0





## 1. Introduction

The decrease in the production costs of renewable energies, as well as the influence of programs such as the 2030 Agenda, have spurred on the generation of electricity based on these sources. One of the fastest-growing sources is solar energy: in 2019 alone, it grew by 22%, which represents an increase of 131 TW/h compared to 2018 [1].

Electricity generation through solar energy is carried out mainly through Photovoltaic Solar Farms (PVSF). These are subject to weather conditions, so their generation window is short. To achieve greater efficiency and reduce the time of affectation due to possible breakdowns, it is necessary to develop systems that, by integrating new technologies with existing systems, allow the development of more robust systems with predictive capacity.

A fundamental part of the PVSF, and one that is most prone to breakdowns, is the Inverter component. This is the equipment in charge of transforming the energy obtained through the solar panels from direct current into alternating current available for transmission and distribution.

Currently, thanks to technological advances such as Industry 4.0 and the Internet of Things (IoT) and the development and popularization of techniques in this framework such as Digital Twins (DT), they have achieved an increase in efficiency, security, and industry sustainability.

The objective of this research is to obtain a DT of a PSFV using an autoencoder architecture, which has the aim of detecting anomalies in the system. For this purpose, we conducted a study over the time series of the PSFV through an analysis of the data, propose an autoencoder architecture for the model of the DT and define an anomaly detection system over it to validate the reliability of the proposed model.

The rest of the paper is organized into three sections. Section 2 describes the main theoretical foundations for solving the stated problem. In Section 3, we present a discussion of related work. In Section 4, we describe aspects of the analysis and development of the

work are detailed. Section 5 describes a series of experiments is carried out with different data sets and an analysis of the results obtained is carried out. Finally, conclusions and future work are presented.

## 2. Background

In this section, time series are described, as well as some techniques used for their processing. In addition, the DL algorithms used for unsupervised learning are presented, and finally an outline is given of DTs, anomaly detection systems and their application in photovoltaic energy systems.

### 2.1. Time Series

A time series is a succession of data observed in periods of time ordered chronologically. These allow us to study the relationships between variables that change over time and their influence on each other. They can be classified as univariable and multivariable:

**Definition 1.** *A univariate time series $X = \{x_t\}_{t \in T}$ is an ordered sequence of observations, where each observation corresponds to a time $t \in T \subseteq Z^+$ [2].*

**Definition 2.** *A multi-variate time series $X = \{x_t\}_{t \in T}$ is defined as an ordered sequence of k-dimensional vectors, where each vector is observed at a specific time $t \in T \subseteq Z^+$ and consists of k observations, $x_t = \{x_{1t}, \ldots, x_{kt}\}$ [2].*

There are multiple techniques for the analysis of time series to explore and extract important characteristics that allow a better understanding of the time series and identify possible patterns. These techniques can be grouped according to the domain in which they are applied as the time domain or frequency.

#### 2.1.1. Time Domain Analysis

In the time domain, statistical methods are relevant. Characteristics such as the mean, standard deviation and variance provide us with information on the trend, stability and probabilistic distribution of a signal.

Other more sophisticated techniques, such as kurtosis, skewness, and correlation, are used to understand the normality of a signal, as well as the relationships between this and other signals. We briefly detail these techniques as follows:

- Kurtosis provides information on the shape of the distribution [3].
- Skewness indicates the asymmetry of the distribution, with negative values indicating a bias to the left and positive values indicating a bias to the right [3].
- Correlation measures the linear relationship between the signals, and it is widely used to detect highly correlated signals to perform dimensionality reduction [4].

The application of these techniques in combination with smoothing techniques achieves a better representation of the sensor signals, improving the performance of the pattern recognition algorithms. There are different smoothing techniques, including exponential, logarithmic, and exponential decay, among others. The objective of these methods is to clean the signals from noise.

Other techniques that allow us to carry out an in-depth study of time series in the time domain are the seasonal decomposition, auto-correlation function, partial autocorrelation and the autoregressive models, such as moving average, autoregressive moving average and integrated autoregressive moving average. Each of these techniques provides us information about the predictability of the time series, such as the influence of random processes or the linearity and influence of previous values.

#### 2.1.2. Frequency Domain Analysis

Time series are usually signals with a high presence of noise, which are not easy to treat using time-domain techniques. For this reason, an analysis in the frequency domain is

recommended. The Fourier transform allows us to detect noise by decomposing the signal and observe the resulting frequency spectrum.

Some of the techniques that allow analysis in the frequency domain are:

- Amplitude vs. frequency: allows observing the frequency response of a signal, which helps to detect the presence of noise at low or high frequencies.
- Spectrogram: shows the strength of a signal over time of the frequencies in the signal. It allows detecting the presence of harmonics in the signal.

In the frequency domain, the application of a filtering process is widely used to eliminate the presence of noise. Among the most used filters are the Butterworth and Chebyshev [5,6].

### 2.2. Deep Learning

Within the field of DL [7], convolutional networks (CNNs) [8] have demonstrated their effectiveness in multiple problems like object recognition, image segmentation and classifying sequences. Other popular neural network architectures are the recurrent ones, like LSTM (long short-term memory) [9], which are able to learn patterns hidden in time series. Several studies have shown the efficacy of both in the detection of anomalies [10,11].

Another DL family widely used for time series is the autoencoders. These are deep neural networks that use a non-linear reduction of dimensionality to learn the representations of the data in an unsupervised way. In this way, the network can "self-learn" the optimal parameter representation that the data set represents. This functionality has found extensive use in different applications, one of them being anomaly detection [7]. By using normal data for training, the autoencoder can learn the correct representations of a system, minimizing the reconstruction error and, in turn, using it as an anomaly evaluation criterion. As a result, autoencoders are able to detect anomalies in a simple way.

#### 2.2.1. Recurrent and Convolutional Networks

A convolutional neural network is a neural network in which a convolution operation is used instead of matrix multiplication operations in at least one of its layers [7]. These networks, by combining convolutional layers, pooling layers and dense layers, can correctly capture the spatial-temporal interdependencies of the provided samples.

A brief description of the main layers of a CNN is as follows:

- Convolutional Layer: In this layer, a convolution operation is applied to its input by means of a kernel to extract a map of characteristics. It can be denoted as $s(t) = \sum_{\alpha=-\infty}^{\infty} x(\alpha) w(t - \alpha)$ where $x$ is the input and $w$ the kernel, and the output can be referred to as a feature map. It presents as main parameters the number of filters to apply and the size of the kernel.
- Pooling layer: A pooling function is applied by replacing the network output at a certain location with a statistical summary of nearby points. This operation helps to make the representation invariant to small variations in the input.
- Dense or fully connected layer: This layer operates on a flattened input, where each input is connected to all neurons. They are usually used at the end of the convolutional architecture and are used with the aim of optimizing the score of the classes.

Proposed by Hochreiter and Schmidhuber [9], LSTMs avoid gradient fading by including a so-called "state cell", the information of which is carefully regulated by structures called gates. The gates can be denoted as $\Gamma = \sigma(W_{x^t} + U_{a^{t-1}} + b)$, where $W$, $U$, $b$ are specific coefficients of the gate and $\sigma$ is the sigmoid function.

The dependencies between the state $c^t = \Gamma_u \, \widetilde{c}^t + \Gamma_f \, \widetilde{c}^{t-1}$ are such that the vector with the new values of the state cell $\widetilde{c}^t = \tanh(W_c[\Gamma_r \, a^{t-1}, x^t] + b_c$ and activation $a^t$, where $\Gamma_u$ is the update gate, $\Gamma_r$ is the relevance gate, $\Gamma_f$ is the forgetting gate, $\Gamma_o$ is the exit gate and $W_c$, $b_c$ are coefficients of the state cell.

### 2.2.2. Autoencoders

An autoencoder is defined as a neural network trained to replicate a representation of its input to its output [7]. Internally, it consists of two parts: an encoder function $h = f(x)$ that encodes the set of input data in a representation of it, usually of a smaller dimension, and a decoder function $r = g(h)$ that performs the input reconstruction through this representation, trying to decrease the reconstruction error (*RE*):

$$RE(i) = \sqrt{\sum_{j=1}^{D} \left( x_j(i) - \hat{x}_j(i) \right)^2} \tag{1}$$

The learning process can be defined with a loss function to optimize $L(x, g(f(x)))$ [7].

One limitation of autoencoders is that although they achieve a correct representation of the input, they do not learn any useful characteristics from the input data. To avoid this, a regularization method is added to the encoder by modifying the loss function $L(x, g(f(x))) + \Omega(h)$. This allows it not only to decrease the reconstruction error but also to learn useful features.

Autoencoders make it possible to develop an architecture that can reconstruct a reliable representation from an input set and in turn can learn relevant characteristics about it [12]. In our case, they can be used as a basis to obtain the digital representation of a system through the learning of their historical data.

### 2.3. Digital Twins

Multiple definitions of the Digital Twin have emerged since the term was first coined by Grieves [13]. In the study carried out by Fuller [14] and Barricelli [15], several of these definitions are addressed, and through them, we can reach the conclusion that a Digital Twin is nothing more than the digital replica of an object or physical system, constantly evolving through a connection to the physical system.

The development of technologies such as Big Data, IoT and Industry 4.0, as well as the use of AI, have allowed the development of DTs through the implementation of data-based models using both machine and deep learning algorithms [16]. These data-driven models present an alternative where less expert knowledge is required for their implementation.

The use of DL algorithms to obtain DT from a physical system has been widely studied, with autoencoders [17] being one of the most useful. This is mainly due to its ability to build representations based on linear or non-linear relationships, as well as features present in its input.

Detection of anomalies is one of the main DT applications. By comparing the DT data obtained in normal behavior, the DT can determine if a deviation occurs in the physical system that is a symptom of the occurrence of an anomaly [18,19].

### 2.4. Anomaly Detection

From a statistical point of view, we can say that an anomaly is an outlier in a distribution [18]. These can be classified into three types of anomalies: specific, contextual or collective anomalies [19].

When the detection system is supervised, the use of machine learning has proven its effectiveness. This performance of machine learning algorithms declines when it is necessary to capture complex structural relationships in large data sets [18].

For the analysis of multi-variable and high-dimensionality sets, studies such as that of Blázquez-García [2] or that of Chalapathy [18] show that by using deep learning techniques, these complex structural relationships can be captured in data sets with those characteristics. Among these techniques are autoencoders. There are several studies that use autoencoders for anomaly detection [20–23]. However, there are few studies that link its use in obtaining a DT for anomaly detection [15] in PVSF. The study developed by Booyse [17] demonstrates the effectiveness in the diagnosis using deep learning techniques to obtain a DT and its subsequent use in the detection of possible faults.

For the detection of anomalies in sets such as the one described above, three detection techniques are mainly used: sequential, spatial and graphic detection [19]. Sequential detection is applied to sequential data sets (e.g., time series) to determine the sub-sequences that are anomalous [24].

These detection techniques vary in their implementation and criteria according to the application domain. One of the most general methods used to determine an abnormal value or score is by calculating the *RE* [18]. By using a training set with normal samples, the minimum *RE* is obtained, which is used as the anomaly threshold, to obtain the assessment of either the anomaly or its labels.

## 3. Related Work

The study developed by Castellani [25] demonstrates its use in order to obtain the DT to both generate normal data in order to simulate the system to be studied and later use it to detect anomalies.

In the study carried out by Harrou [26], authors apply a model-based approach to generate the characteristics of the photovoltaic system studied. This type of approach presents the drawbacks of the need to have a level of expert knowledge in the photovoltaic system, and it does not take advantage of the availability of information provided by SCADA systems.

Studies based in deep learning methods are gained more attention. Other studies like De Benedetti [27] or Pereira [28], authors present approaches based on Deep Learning models (the first based on Artificial Neural Networks and the second on Variational Recurrent Autoencoders), with the aim of obtaining models to detect anomalies in photovoltaic systems. These studies obtained good results but are limited to uni-variable series, which does not capture all the possible relationships between the signals of the system to be studied.

Jain [20] carried out a study of the use of a DT for fault diagnosis in a PVSF but through a mathematical modeling. Unfortunately, there are very few studies that carry out an analysis using a data-based approach.

In our study we intend to carry out the approach from a multi-variable perspective in order to be able to obtain the DT of a PVSF inverter using an autoencoder architecture by combining convolutional and recurrent layers to better capture the relationships between the time series obtained from the inverter. The contributions of this paper are summarized as follows:

- Obtain a DT from a PVSF using a data driven approach.
- Using the obtained DT for anomaly detection using two different methods.

## 4. Methodology

This section details the entire analysis and implementation process that has been carried out in this paper. First, the data set was extracted from the Sunny Central 1000CP XT Inverter, manufacturer CR Technology Systems, Trevligio, Italy. Next, pre-processing was done to decrease the dimensionality. Subsequently, an analysis of characteristics in the time and frequency domains was carried out to study their possible impact on the performance of the proposed model. To obtain the DT, the use of an autoencoder is proposed. Finally, the reconstruction error of the digital twin obtained is used as a reference for the anomaly detection system.

### 4.1. Exploratory and Data Analysis

The starting data set was obtained from an Inverter of the SMA firm model SC-1000CP-XT in the period between 1 July 2019 until 16 March 2020 and consists of a daily CSV file with 288 measurements of 259 different metrics.

The following pre-processing tasks were performed on this data set:

- We removed all categorical variables from data sets.
- We identified the variables that establish the temporal index of the set.
- We selected data only from the time slot corresponding to the incidence of sunlight.

- We used the Kendall method to study the correlation between the different signals, eliminating highly correlated signals (with an index greater than 0.9), with the aim of reducing the dimensionality of the set.
- We studied by subsets, dividing the main signals (those that are direct inputs or outputs to the Inverter) into a subset and the rest, coinciding with the String currents, which are separated depending on their Strings monitor (8 signals per monitor, 14 monitors total).

We determined from this analysis that only the variables of the main subset provided valuable information. Thus, the final data set only includes 10 metrics and 120 daily measurements of each of them.

### 4.2. Time Domain Analysis

For the analysis in the time domain, the following four characteristics were initially selected for the analysis: the mean, variance, kurtosis and skewness, to determine the distribution of the data set, its symmetry and presence of bias, detecting a high dispersion in the measurements of the signals, as well as a strong presence of bias in them. To improve the distribution of the set, a logarithmic smoothing was carried out.

Subsequently, the Augmented Dickey-Fuller test was performed to check the seasonality of the signals, obtaining a $p$-value equal to zero. To verify the existence of patterns in the different signals, Autocorrelation Function (ACF) and Partial Autocorrelation Function (PACF) were used, observing the presence of a predictability pattern as seen in Figure 1 in one of the metrics selected for the study.

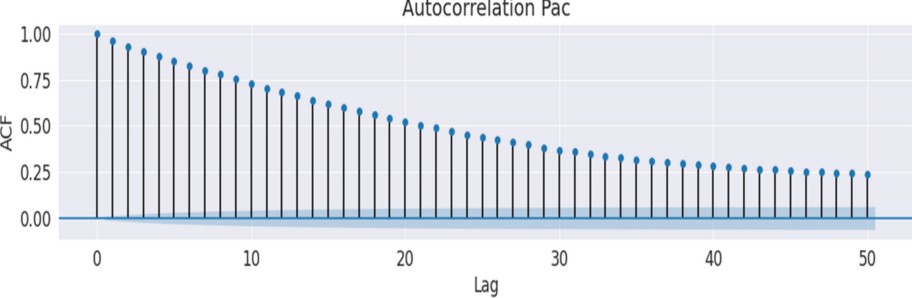

**Figure 1.** Autocorrelation in Power AC (Pac).

### 4.3. Frequency Domain Analysis

To perform the analysis in the frequency domain, initially, the Fourier transform of the data set was applied. An examination of the frequency distribution of the signals was carried out, choosing one day as the time interval for the study. In Figure 2, the normality of the signal in the frequency of the Pac signal is shown; for a better visualization, several percentiles (5, 10 and 15) were defined.

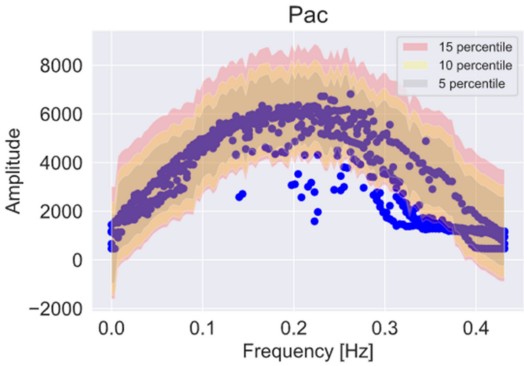

**Figure 2.** Normality study in Power AC (Pac).

To check the effect of possible noise, the signals were processed, applying a filtering process using the Butterworth method and using two filtering processes to check its incidence on the signals, one using a low-pass filter and the other using a filter passband. We obtained a "cleaner" signal in the case of the low-pass filter. By using this filtering process, a new set with the signals filtered with a low-pass filter was generated.

### 4.4. Architecture of the Digital Twin

To obtain the DT, we propose the use of an autoencoder architecture. This choice is due to characteristics of this type of architecture to encode the signals, which allows a better understanding of the relationships between the signals to obtain the model. For this, a combination of different layers was used in the architecture as follows:

- Encoder:

  - Convolutional Conv1D Layer: the objective of this layer is to extract the most significant characteristics from the data set.
  - Convolutional Conv1dTranspose Layer: the objective of this layer is to return to the configuration of the input, keeping the characteristics learned by the previous layer.
  - LSTM Recurrent Layer: By using this layer, it is intended that the network can learn the temporal relationships that are established between a multivariate data set.
  - RepeatVector reorganization layer: This layer is used as an adapter that allows the Encoder output to be concatenated with the Decoder input.

- Decoder:

  - LSTM Recurrent Layer: By using this layer, it is intended that the network can decode the temporal relationships established between a multivariate data set.
  - TimeDistributed Recurring Layer: This layer is used as a container for a Dense layer so that each weight can produce an output at each instant of time (time step).

The combination of Conv1D and LSTM layers allows mixing characteristics of the time series (Conv1D) and the time relationships between the different series. Table 1 shows the characteristics of the proposed autoencoder.

**Table 1.** Characteristics of the Autoencoder.

|  | **Layers** | **Characteristics** |
|---|---|---|
|  | Input | Shape: time steps x metrics |
| Encoder | Convolutional1D | Number of filters: 64<br>Kernel size: 7<br>Regularization function: L2 = 0.0001<br>Activation function: 'relu' |
|  | Dropout | Frequency: 0.2 |
|  | Transpose Conv1D | Number of filters: metrics<br>Kernel size: 7 |
|  | LSTM | Number of layers: 2<br>Number of output units: 128 and 64<br>Activation function: 'relu' |
|  | Repeat Vector | Repetition factor: metrics |
| Decoder | LSTM | Number of layers: 2<br>Number of output units: 64 and 128<br>Activation function: 'relu' and 'softmax' |
|  | Time Distributed | Layer: Dense (Number of Units: metrics) |

By using the DT, it is proposed to obtain the reconstruction of the system signals, as well as its reconstruction error, to be able to establish using this reconstruction error as a threshold for detecting anomalies.

### 4.5. Algorithms for Anomaly Detection

In this paper, we propose two methodologies based on the detection of anomalies using the *RE*. The first method (or local detection) captures the residual deviation of the model obtained with respect to the real system, establishing a threshold for labeling the anomaly between mild or serious. The second method (fine detection) checks the behavior in a period in order to establish if the supposed anomaly could be just an error in the measurement or if we are in the presence of an anomalous behavior (persistence of the anomaly in a period of time). They are briefly detailed below:

- First method: A set of thresholds are established using as a criterion the degree of deviation $\delta$ up to a 15% variation with respect to the *RE*:

  - If $\delta > RE \rightarrow$ slight anomaly.
  - $0.1\delta > RE \rightarrow$ - If $0.15\delta > RE \rightarrow$ serious anomaly.

- Second method: The *RE* is defined as threshold, and the following criteria are established for labeling the anomaly:

  - If at least 80% of the signals at the same instant of time $t$ show $\delta > RE$.
  - If the deviation $\delta$ hasbeen repeated in at least $\pm 5t \rightarrow$ serious anomaly.

The steps implemented for the anomaly detection system are described below:

1. Obtain the reconstruction set, $\hat{X}_t = \{\hat{x}_{1t}, \ldots, \hat{x}_{kt}\}$ with the prediction of our model using the training $X_t = \{x_{1t}, \ldots, x_{kt}\}$.
2. With this reconstruction set, the *ER* reconstruction error on the training set is obtained.
3. Obtain the reconstruction set, $\hat{X}_t = \{\hat{x}_{1t}, \ldots, \hat{x}_{kt}\}$ with the prediction of our model using the test set $X_t = \{x_{1t}, \ldots, x_{kt}\}$.
4. On the set $\hat{X}_t$, apply the first method.
5. On the set $\hat{X}_t$, apply the second method.

Subsequently, an evaluation of the effectiveness in detecting anomalies of the models obtained through the different processing techniques was carried out. For this purpose, a set of manually labeled samples was used according to the criteria of experts, and using metrics such as precision, recall or AUC, the proposed methods were evaluated.

## 5. Experiments and Results

This section shows and analyzes the results of the experiments carried out. Initially, the characteristics of the subsets resulting from the application of both the pre-processing stages and the analysis in the time and frequency domain are shown. Subsequently, their impact on obtaining the DT is analyzed, as well as validated in the detection of anomalies. In Table 2 we show the time intervals chosen for the training, validation and test sets.

**Table 2.** Characteristics of the data sets.

|  | **Periods** | **Samples** |
|---|---|---|
| Training | "10 October 2019": "17 December 2019" | 8349 |
| Validation | "27 August 2019": "27 September 2019" | 3872 |
| Test | "21 December 2019": "15 March 2020" | 10,406 |

### 5.1. Experiment Design

The design of experiments was divided into two stages: a first stage, where four different data sets were obtained by applying data cleaning techniques, time domain, frequency domain and a combination of all the above (as seen in Table 3). The DT was applied to these different data sets in order to check the effect of processing techniques on

DT performance. Subsequently, the anomaly detection process was carried out to verify the effectiveness of the DT and the effect of the processing techniques on it, with the DT being selected that presents the best anomaly detection score. In the second stage, we established a comparison between three DL methods and one auto-regressor method in order to evaluate the performance of the proposed DT.

**Table 3.** Applied processing techniques.

|  | **Techniques** |
| --- | --- |
| Pre-processing | Data cleaning |
|  | Reduction of dimensionality |
| Time | Logarithmic Smoothing |
|  | Seasonal Decomposition |
| Frequency | Low-pass filter |
| Time-Frequency Combination | Logarithmic Smoothing |
|  | Seasonal Decomposition |
|  | Low-pass filter |

*5.2. Results*

To carry out the evaluation of our model in the detection of anomalies, a time interval of seven days was selected, from 7 February 2020 to 13 February 2020. Manual labeling of the anomalous samples was performed according to the experts. Table 4 shows the results for the proposed model. In the two first rows, we present the results obtained by the proposed model based on the loss and the *RE*. In the last rows, we present the results in the detection of anomalies using the methods previously described. It can be noticed that, despite having a lower loss and reconstruction error, the corresponding DT performance declines in terms of anomaly detection. This is because the more intensive the data processing is, the more prone to discard features that can help the model to learn better the physical system.

**Table 4.** DT performance in different parts and domains (cells in bold report best scores).

|  |  | **Pre-Processing** | **Time** | **Freq.** | **Time/Freq.** |
| --- | --- | --- | --- | --- | --- |
| Loss | Training | 0.0061 | 0.0089 | 0.0020 | 0.0021 |
|  | Test | 0.0196 | 0.0374 | 0.0047 | 0.0036 |
| RE | Training | 0.0441 | 0.0698 | 0.0362 | 0.0295 |
|  | Test | 0.1074 | 0.1707 | 0.0588 | 0.0439 |
| F. Mth. | Precision | 0.38 | 0.19 | 0.04 | 0.08 |
|  | Recall | 0.83 | 0.76 | **0.73** | **0.69** |
|  | AUC | 0.89 | 0.83 | **0.63** | **0.72** |
| S. Mth. | Precision | 0.53 | 0.39 | - | - |
|  | Recall | **0.92** | **0.92** | - | - |
|  | AUC | **0.97** | **0.93** | - | - |

In the second stage, it was decided to make the comparison of our model in the data set only with pre-processing, against the following configurations: CNN autoencoder (A_CNN), with two convolutional layers (Conv1D) in the encoder and two layers transposed from the convolutional (Conv1DTranspose) in the decoder; LSTM autoencoder (A_LSTM), two LSTM layers in the encoder and two LSTM layers in the decoder; Simple LSTM (S_LSTM), a single LSTM layer with L2 = 0.0001, and a Dense layer at the output. Finally, the Vector Auto Regression (VAR) method was used. Results of these samples are shown in Tables 5 and 6. For the evaluation of the first method, all anomalies were selected.

**Table 5.** Number of automatically detected anomalies.

|  |  | 7 February to 13 February |
|---|---|---|
| **Total Samples** | | **840** |
| Anomalies detected by applying the first method | DT | 380 |
| | A_CNN | 221 |
| | A_LSTM | 442 |
| | S_LSTM | 513 |
| | VAR | 201 |
| Anomalies detected by applying the second method | DT | 45 |
| | A_CNN | 39 |
| | A_LSTM | 24 |
| | S_LSTM | 18 |
| | VAR | 0 |
| Manually tagged anomalies | | 26 |

**Table 6.** Comparison results of between different algorithms (cells in bold report best scores).

|  |  | DT | A_CNN | A_LSTM | S_LSTM | VAR |
|---|---|---|---|---|---|---|
| F. Mth. | Precision | 0.38 | 0.09 | 0.04 | 0.03 | 0.09 |
| | Recall | 0.83 | 0.76 | 0.80 | 0.61 | 0.73 |
| | AUC | 0.89 | 0.75 | 0.64 | 0.49 | 0.75 |
| S. Mth. | Precision | 0.53 | 0.51 | **0.87** | **0.88** | INF |
| | Recall | **0.92** | 0.76 | 0.80 | 0.61 | 0 |
| | AUC | **0.97** | 0.86 | 0.89 | 0.80 | 0.5 |

As can be seen, the proposed DT model obtains the best recall and AUC values, although it presents a lower precision in the second anomaly detection method compared to models using only LSTM (A_LSTM and S_LSTM). Another point to highlight is that the model presents a consistent and better performance in both methods. Conversely, VAR completely fails in detecting anomalies for the second method.

As can be seen in Figure 3, the proposed model presents a more robust response to the anomalies detected by the second proposed method, despite the possible influence of weather conditions. This gives us clues about including the climatic variables in the model for improving its detection performance.

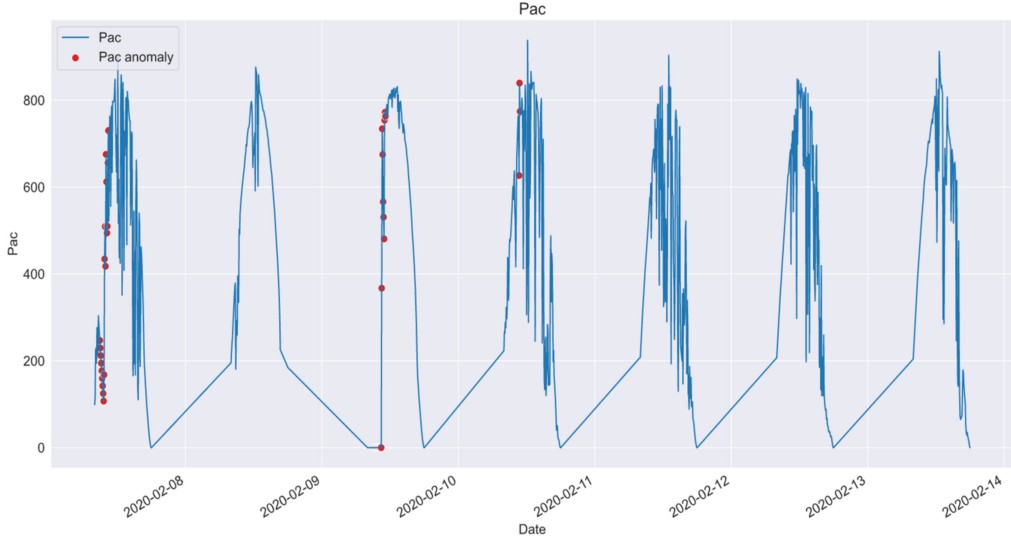

**Figure 3.** Anomalies detected by the proposed model using the second method in Power AC (Pac).

## 6. Conclusions and Future Work

In the present paper, a procedure was developed to obtain the Digital Twin of a system through the analysis of time series and through the application of Deep Learning techniques and using it to detect anomalies. For the solution of the objective, a pre-processing and analysis of characteristics was carried out first, carrying out a study of their influence on the studied signals, and later analyzing their influence on the proposed task.

To obtain the Digital Twin, an approach based on deep learning using autoencoders was used, thereby combining convolutional layers and LSTM, as well as the influence of different parameters and the objective of obtaining the Digital Twin with a reconstruction error close to 0.1.

Subsequently, several experiments were carried out to determine the influence of the pre-processing of the signals both in obtaining the DT and in the subsequent detection of anomalies. It was observed that the application of processing techniques in both frequency and time domains improves the response of the model, reducing the loss function and the reconstruction error. However, the application of these techniques has two effects depending on the method used to detect anomalies: one, causing the model to be more susceptible to non-anomalous oscillations in the signals; two, the model does not detect any anomaly.

Considering these results, several ideas have been raised for future work to carry out, as described below:

- We plan to connect the DT to its physical system during periods with special conditions so that the DT can learn more specific situations that can mislead the detection of anomalies. This connection can be also useful to teach the DT how to simulate some parameters of the physical system and the earlier detection of potential failures.
- One limitation of this study is the lack of meteorological variables, which provide vital information for the analysis and better understanding of the temporal relationships established between the different signals. In future work, we will perform further experimentation by including these variables when predicting anomalies.

**Author Contributions:** Conceptualization, K.A. and R.B.; methodology, K.A. and R.B.; software, K.A.; validation, K.A. and R.B.; formal analysis, K.A. and R.B.; investigation, K.A. and R.B.; resources, K.A. and R.B.; data curation, K.A.; writing—original draft preparation, K.A.; writing—review and editing, R.B.; visualization, K.A.; supervision, R.B.; project administration, K.A. and R.B. All authors have read and agreed to the published version of the manuscript.

**Funding:** This project has been funded by the Ministry of Economy and Commerce with project contract TIN2016-88835-RET and by the Universitat Jaume I with project contract UJI-B2020-15.

**Institutional Review Board Statement:** Not applicable.

**Informed Consent Statement:** Not applicable.

**Data Availability Statement:** Data used in the experiments are publicly available through the following URL: https://krono.act.uji.es/DigitalTwins/PSFV_dataset.txt (accessed on 21 March 2021).

**Conflicts of Interest:** The authors declare no conflict of interest.

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
