# Peer review of "Digital Twins in Solar Farms: An Approach through Time Series and Deep Learning"

_algorithms, doi:10.3390/a14050156_

Round 1

Reviewer 1 Report

The authors present, a procedure was developed to obtain the Digital Twin of a system through the analysis of Time Series and through the application of Deep Learning techniques and use it to detect anomalies. The issue of the article is very interesting and is in the scope of the magazine. In addition, it is in the current state of the art.

-After a fairly lengthy introduction, the contribution in the last paragraph is not entirely clear. Please, explain more this, if it is possible.

- In the same way, in order to improve understanding, it is important to separate the last paragraph into two: contribution on the one hand and subsections on the other.

- The equation “ER” is not numbered. Please, modify this.

- In the results section, the results are displayed only in tables. It would be very interesting if this result is was shown in a figure to improve the compression.

- The provided argument in the introduction and literature review section seems weak since the found opportunity is addressed in a general way. The authors must deepen the clarification of the problem and its relevance. Techniques based on artificial intelligence are not novel but nevertheless, it is possible that they are in the particular application in which the work is focused. Perhaps it is important to reference works that these anomaly detections are used in other areas and are not so explored in the particular case of this work. For example

Castaño, F., Haber, R.E., Mohammed, W.M., Nejman, M., Villalonga, A., Martinez Lastra, J.L.; Quality monitoring of complex manufacturing systems on the basis of model driven approach (2020) Smart Structures and Systems, 26 (4), pp. 495-506. DOI: 10.12989/sss.2020.26.4.495

Haber, R., Strzelczak, S., Miljkovic, Z., Castano, F., Fumagalli, L., Petrovic, M.; Digital twin-based Optimization on the basis of Grey Wolf Method. A Case Study on Motion Control Systems (2020) Proceedings - 2020 IEEE Conference on Industrial Cyberphysical Systems, ICPS 2020, pp. 469-474. DOI: 10.1109/ICPS48405.2020.9274728

Author Response

Response to Reviewer 1 Comments

Point 1: The authors present, a procedure was developed to obtain the Digital Twin of a system through the analysis of Time Series and through the application of Deep Learning techniques and use it to detect anomalies. The issue of the article is very interesting and is in the scope of the magazine. In addition, it is in the current state of the art.

Response 1: We thank the reviewer for his/her very nice comments on our study.

Point 2: After a fairly lengthy introduction, the contribution in the last paragraph is not entirely clear. Please, explain more this if it is possible.

Response 2: We appreciate and thank the reviewer for his/her recommendations. A new paragraph have been add with a more clear contribution.

Point 3: In the same way, in order to improve understanding, it is important to separate the last paragraph into two: contribution on the one hand and subsections on the other.

Response 3: We appreciate and thank the reviewer for his/her recommendations. We have separate the contribution and the subsections into two different paragraphs.

Point 4: The equation “ER” is not numbered. Please, modify this.

Response 4: We have added a number to equation “ER” ( see line 224).

Point 5: In the results section, the results are displayed only in tables. It would be very interesting if this result is was shown in a figure to improve the compression.

Response 5: We appreciate and thank the reviewer for his/her recommendations. We have added a figure with the results of the DT for anomaly detection.

Point 6: The provided argument in the introduction and literature review section seems weak since the found opportunity is addressed in a general way. The authors must deepen the clarification of the problem and its relevance. Techniques based on artificial intelligence are not novel but nevertheless, it is possible that they are in the particular application in which the work is focused. Perhaps it is important to reference works that these anomaly detections are used in other areas and are not so explored in the particular case of this work.

Response 6: We appreciate and thank the reviewer for his/her recommendations. We have modified the structure and added more references to related work in order to clarify the relevance of our study.

Reviewer 2 Report

The work under review is interesting and very well-written. It looks like a book rather than like a research paper in its first 6-7 pages. The idea to combine deep learning and digital twin technology is not novel. However, the implementation steps demonstrate a good approach and will be interesting for the audience. I have few comments aiming to improve the quality of the description as:

  • All abbreviations should be defined separately in the abstract and in the main text at their first occurrence regardless being widely used and known. For example, look at AUC, PSFV (or PVSP), ACF, PACF, A_CNN, A_LSTM, S_LSTM, VAR.
  • Use singular instead of plural on line 75.
  • Pay attention to the Figures 1, 2 and 3 as written approvals from the authors are necessary to be included in the paper.
  • Definition of ER is on page 6 and first use on page 5.
  • There are no dimensions on the axes in Figures 4 and 5. In fact there are two Figures 4.
  • Figure 4 presenting frequency domain is not readable in black and white.
  • Use on 0.1*delta in line 438 is pointless.
  • Text explaining results on Table 4 is not clear.

Author Response

Response to Reviewer 2 Comments

Point 1: The work under review is interesting and very well-written. It looks like a book rather than like a research paper in its first 6-7 pages. The idea to combine deep learning and digital twin technology is not novel. However, the implementation steps demonstrate a good approach and will be interesting for the audience. 

Response 1: We thank the reviewer for his/her very nice comments on our study.

Point 2: All abbreviations should be defined separately in the abstract and in the main text at their first occurrence regardless being widely used and known. For example, look at AUC, PSFV (or PVSP), ACF, PACF, A_CNN, A_LSTM, S_LSTM, VAR.

Response 2: We appreciate and thank the reviewer for his/her recommendations. We have added all definitions of abbreviations used in the study. 

Point 3: Use singular instead of plural on line 75.

Response 3: We have fixed this error (see line 83). 

Point 4: Pay attention to the Figures 1, 2 and 3 as written approvals from the authors are necessary to be included in the paper.

Response 4: We have concluded that these figures did not provide any add value to the paper so we have decided to eliminate them. 

Point 5: Definition of ER is on page 6 and first use on page 5.

Response 5: We have corrected this issue (see the line 224). 

Point 6: There are no dimensions on the axes in Figures 4 and 5. In fact there are two Figures 4.

Response 6: We have corrected this issue see Figures 1 and 2. 

Point 7: Figure 4 presenting frequency domain is not readable in black and white.

Response 7: We have corrected this issue. 

Point 8: Use on 0.1*delta in line 438 is pointless.

Response 8: We have corrected this issue. 

Point 9: Text explaining results on Table 4 is not clear.

Response 9: We have added a new line for a better explanation of Table 4. See lines from 514 to 516.

Reviewer 3 Report

This work used a deep learning approach to obtain digital twin of a Photovoltaic Solar Farm.

I think Authors need to do much more to get publish this work.

  •  References 23 to 36 are not used in the main text. why?
  • The main contribution/novelty is not mentioned mentioned clearly. This should be included in introduction section.
  • Previously published relevant work is not included in section 2, while authors named this section 'Related work'.
  • There is no discussion on the used approach with previously published approaches
  • The used approach and experiment is also not clearly described
  • Introduction section does not cover the area of research in journal, the summery of the proposed approach and the authors contribution.

Author Response

Response to Reviewer 3 Comments

Point 1: References 23 to 36 are not used in the main text. why?

Response 1: We thank the reviewer for noticing this issue. Most of these references are part of the related work and others where imported wrongly to the document. We have properly cleaned and arranged the references in the new version.

Point 2: The main contribution/novelty is not mentioned clearly. This should be included in introduction section

Response 2: We agree with the recommendation. We have slightly rewritten the introduction to explain more clearly the contribution of the paper. See lines from 40 to 44. 

Point 3: Previously published relevant work is not included in section 2, while authors named this section 'Related work'

Response 3: We have included a "Related work" section, restructuring parts of the paper. We have also included several references to previous relevant work, which were previously included in the bibliography. 

Point 4: There is no discussion on the used approach with previously published approaches.

Response 4: As mentioned earlier, we have included the references of previously published work and we have discussed them in the paper. 

Point 5: The used approach and experiment is also not clearly described.

Response 5: We have rewritten some parts of the paper in order to make it clearer. 

Point 6: Introduction section does not cover the area of research in journal, the summery of the proposed approach and the authors contribution.

Response 6: We have corrected this issue.

Round 2

Reviewer 1 Report

The authors have modified the article according to the reviewer´s comments. For this, the scientific soundness and the quality of presentation have improved.

Author Response

Response to Reviewer 1 Comments

Point 1: The authors have modified the article according to the reviewer´s comments. For this, the scientific soundness and the quality of presentation have improved.

Response 1: We thank the reviewer for his/her very nice comments on our study.

Reviewer 3 Report

Authors response is not satisfactory and I still think this paper needs improvements (which I mentioned previously) before possible publication. 

Author Response

Response to Reviewer 3 Comments

Point 1: Authors response is not satisfactory and I still think this paper needs improvements (which I mentioned previously) before possible publication. 

Response 1: We thank the reviewer for noticing this issues. We have properly restructuring parts of the paper. We have also included several references to previous relevant work, which were previously included in the bibliography.

Round 3

Reviewer 3 Report

I have again reviewed the manuscript with reference to comments of my first review.

I don't understand why Authors have provided defination /background stuff under related work section. The cited work without any further discussion does not make sense. The heading of section 3 and the contents does not match. I would suggest authos to have a look on the published related literature and then re-consider the structure of manuscript accordingly.